

# LSPConv: local spatial projection convolution for point cloud analysis

Haoming Zhang[1], Ke Wang[1,2], Chen Zhong[3], Kaijie Yun[1], Zilong Wang[1], Yifan Yang[1] and Xianshui Tao[4]

[1] State Key Laboratory of Robotics and Systems, Harbin Institute of Technology, Harbin, China
[2] Zhengzhou Research Institute, Harbin Institute of Technology, Zhengzhou, Henan, China
[3] Shenyang Fire Science and Technology Research Institute of MEM, Shenyang, China
[4] Wuhu Hit Robot Technology Research Institute Co., Ltd., Wuhu, Anhui, China

## ABSTRACT

This study introduces a novel approach, Local Spatial Projection Convolution (LSPConv), for point cloud classification and semantic segmentation. Unlike conventional methods utilizing relative coordinates for local geometric information, our motivation stems from the inadequacy of existing techniques for representing the intricate spatial organization of unconsolidated and irregular 3D point clouds. To address this limitation, we propose a Local Spatial Projection Module utilizing a vector projection strategy, designed to capture comprehensive local spatial information more effectively. Moreover, recent studies emphasize the importance of anisotropic kernels for point cloud feature extraction, considering the distinct contributions of individual neighboring points. To cater to this requirement, we introduce the Feature Weight Assignment (FWA) Module to assign weights to neighboring points, enhancing the anisotropy crucial for accurate feature extraction. Additionally, we introduce an Anisotropic Relative Feature Encoding Module that adaptively encodes points based on their relative features, further amplifying the anisotropic characteristics. Our approaches achieve remarkable results for point cloud classification and segmentation in several benchmark datasets based on extensive qualitative and quantitative evaluation.

## INTRODUCTION

With the development of 3D scanning technology and related sensors such as depth cameras and vehicle-mounted LIDAR, 3D point cloud data has been playing an irreplaceable role in many fields, such as robot navigation (*Wang et al., 2021*) and autonomous cars (*Liang et al., 2018*; *Wang et al., 2018*; *Shi et al., 2020*; *Deng et al., 2021*; *Liang et al., 2022*; *Li et al., 2022*; *Liu et al., 2023*; *Yang et al., 2023*; *Chen et al., 2023b*). Compared with images, 3D point cloud data can represent the location, shape, and size of objects in space more accurately, which means a stronger ability for spatial description. However, analyzing 3D information from point clouds using deep learning techniques is more challenging than that from 2D images due to its sparse, irregular, and unordered structure. 2D convolution only applies to structured data, such as images. Because of these characteristics, 2D convolution applied to dense structured image information cannot be

Corresponding author
Ke Wang, wangke@hit.edu.cn

directly applied to 3D point clouds to process it To address this issue, previous work has been studied along two general lines. One transforms 3D point clouds into regular data, such as projections onto 2D images (*Su et al., 2015*; *Qi et al., 2016*) or convert into 3D voxels (*Maturana & Scherer, 2015*; *Song et al., 2017*; *Riegler, Osman Ulusoy & Geiger, 2017*), and use 2D or 3D convolution to extract features. This type of processing results in spatial information being lost or repeatedly represented. Voxelization evenly divides the space of the point cloud into several voxels of the same size, so there will be a case that multiple points may fall on the same voxel, resulting in information loss. If converting a point cloud into multiple views, some parts of the point cloud will appear in different views, resulting in duplication of information. Such methods are not conducive to objectively reflecting the information of the original object. The other process point cloud data directly. As a pioneer, PointNet (*Qi et al., 2017a*) utilizes a shared weight multilayer perceptron (MLP) to encode each point and extract global information. To extract fine-grained point cloud local features, based on PointNet, the subsequent works (*Qi et al., 2017b*; *Wang, Samari & Siddiqi, 2018*; *Wang et al., 2019b*; *Zhao et al., 2019*; *Komarichev, Zhong & Hua, 2019*; *Zhao et al., 2021*; *Xu et al., 2021*; *Xiang et al., 2021*; *Wu et al., 2022*; *Qian et al., 2022*; *Ma et al., 2022*; *Lai et al., 2022*; *Wu et al., 2023*; *Park et al., 2023*; *Robert, Raguet & Landrieu, 2023*) suggest different grouping strategies for point cloud local feature extraction, which significantly improves the ability of point cloud analysis.

However, we have found that previous methods often ignore the appropriate modeling of local geometric feature representations in point clouds. On the level of geometric input features, previous grouping strategies typically use the relative coordinates of neighboring points around the central query point to represent the spatial information of a group of local points. We argue that the local spatial information is not adequately represented because such input data has an unconsolidated structure and lacks a comprehensive description of spatial organization. To this end, we aim to devise a module with the purpose of enhancing the local information representation of point clouds. Thus, a Local Spatial Projection module is introduced by utilizing vector projection strategy and regularization techniques to encode the whole region into a space shape feature and regard the obtained space shape feature as the intrinsic information of the point cloud. We employ this approach to obtain appropriate geometric feature representations for the point cloud.

On the level of the strategy of local feature extraction, early approaches (*Qi et al., 2017a, 2017b*) commonly apply various homogeneous convolution kernels, in which all points are processed by the same MLP, ignoring the individual contributions of different neighbor points for the query point. Follow-up works' adaptive/anisotropic kernels (*Xu et al., 2018*; *Thomas et al., 2019*; *Zhao et al., 2019, 2021*; *Xu et al., 2021*; *Zhou et al., 2021*; *Ma et al., 2022*; *Wu et al., 2022*; *Deng et al., 2023*; *Park et al., 2023*) designed according to the corresponding relationship between the neighbor points and the center query point demonstrate impressive local feature extraction capability. Therefore, we inherit this design philosophy and construct anisotropic point cloud local feature extraction operators. Our first main contribution is developing a radial basis function Feature Weight Assignment (FWA) Module that adaptively assigns weights based on the relative distance

of neighboring points. Furthermore, to obtain a qualified local feature extraction kernel, an Anisotropic Relative Feature Encoding Module is introduced. It encodes points' features adaptively according to the relative coordinates and relative features. By incorporating such a design, the convolution kernel shows more emphatic anisotropy when operating on local regions, which automatically analyzes the variability of contributions due to geometrical and feature distinctions. To sum up, we propose a novel operation called Local Spatial Projection Convolution, namely LSPConv. As opposed to the methods based on large-scale multilayer perceptrons and complexly designed structures for weight obtaining that consume massive amounts of memory and computing resources, LSPConv is lightweight and more efficient.

We propose LSPConv as a way to achieve improved accuracy for 3D classification and segmentation tasks. The proposed network architecture has the following contributions:

- Local Spatial Projection Module is constructed to model local spatial information about the input point cloud, improving the network's ability to capture the local geometry.
- LSPConv employs a Radial basis function Feature Weight Assignment Module to dynamically assign weights to each neighboring point, encoding local neighbor points anisotropically.
- In LSPConv, an adaptive operator called Anisotropic Relative Feature Encoding Module is introduced, which encodes points adaptively according to the relative feature.

## RELATED WORK

**Voxelization-based and multi-view methods.** To apply powerful CNNs in 2D vision to 3D point cloud analysis, there have been some strategies (*Su et al., 2015*; *Qi et al., 2016*; *Kanezaki, Matsushita & Nishida, 2018*; *Maturana & Scherer, 2015*; *Wang et al., 2017*; *Song et al., 2017*; *Riegler, Osman Ulusoy & Geiger, 2017*) representing point cloud by voxelization or multi-view pictures. However, the method has two great problems which are information loss and enormous computation costs. To solve the issue, OctNet (*Riegler, Osman Ulusoy & Geiger, 2017*) and Kd-Net attempt (*Klokov & Lempitsky, 2017*) to use more efficient data structures to cut down the cost and extract effective information as much as possible. However, none of these methods can achieve the desired result because it is fundamentally difficult to represent 3D information with one-sided 2D information, especially in the case of large-scale scanning.

**Point-based methods.** Researchers have developed deep network structures that manipulate raw point clouds directly, as sets embedded in continuous space, instead of projecting or quantifying irregular point clouds onto regular grids in 2D or 3D. PointNet (*Qi et al., 2017a*) is a novel deep net architecture suitable for the irregularity of point clouds, which does not rely on the intermediate representation. Due to the disorder and rotation invariance of the point cloud, it rotates the point cloud to a proper pose, and then utilizes MLP on each point to map the information of the point clouds to higher dimensions. Then, it applies a symmetric function to aggregate global features. However, PointNet only considers global features and neglects the capture of local features. PointNet

++ (*Qi et al., 2017b*) has been proposed to solve the issue by applying a local aggregator to capture local correlations, which is called set abstraction (SA) in different scales. PointNet and PointNet++ introduced a groundbreaking paradigm for point cloud analysis, paving the way for subsequent methods. These methods adopt a similar network framework structure, incorporating diverse local feature aggregation modules to enhance model performance. They achieve this by leveraging graph neural networks (*Verma, Boyer & Verbeek, 2018*; *Wang, Samari & Siddiqi, 2018*; *Li et al., 2018a*; *Lei, Akhtar & Mian, 2020*; *Wang et al., 2019b*, *2019a*; *Lin, Huang & Wang, 2020*), introducing meticulously designed kernels (*Xu et al., 2018*; *Thomas et al., 2019*; *Zhao et al., 2019*; *Hu et al., 2020*; *Xu et al., 2021*; *Qian et al., 2022*; *Ma et al., 2022*), or employing attention mechanisms (*Zhao et al., 2021*; *Wu et al., 2022*; *Lai et al., 2022*; *Guo et al., 2021*; *Wu et al., 2023*; *Park et al., 2023*; *Robert, Raguet & Landrieu, 2023*).

**Anisotropic kernel design.** A fixed convolution kernel is limited to detecting the most relevant section in the neighborhood, so a qualified method should generate adaptive kernels instead of the aforementioned isotropic kernels such as a fixed MLP applied in PointNet/PointNet++. Many studies (*Velicković et al., 2017*; *Verma, Boyer & Verbeek, 2018*; *Simonovsky & Komodakis, 2017*; *Wu, Qi & Fuxin, 2019*; *Wang et al., 2019a*; *Lin, Huang & Wang, 2020*) have been proposed to assign proper attentional weights to different points or filters, which are defined as anisotropic kernels as opposed to isotropic kernels. The study of the asymmetric anisotropic kernel has brought enlightenment to this article. A typical example is KPConv (*Thomas et al., 2019*). KPConv updates the feature of kernel points by setting kernel points with different weights and modifying the locations of kernel points on the basis of deformable convolution. PAConv (*Xu et al., 2021*) designs a Weight Bank and a ScoreNet to learn a score based on the position relationship of points and use the score to assemble weight matrices for each point adaptively. By introducing these two modules, the encoding process of each neighbor point can exhibit independence and anisotropy. CurveNet (*Xiang et al., 2021*) introduces an operator to consolidate hypothetical curves within point clouds. Sequences of connected points are grouped through guided traversal of the point cloud. These grouped curves are then further aggregated to enhance their individual point-wise features. The Point Transformer series (*Zhao et al., 2021*; *Wu et al., 2022*) utilize neighborhood attention and updated grouped vector attention to encode each neighbor point individually according to relevant features. PointVector (*Deng et al., 2023*) proposes a vector-oriented point aggregation operator that can aggregate neighboring features through higher-dimensional vectors. SPoTr (*Park et al., 2023*) adopts local points attention (LPA) defined on a local group to learn local shape context, which applies Channel-Wise Point Attention (CWPA) to make the features of each point exhibit anisotropy.

**Methods for rotational invariance.** Recent advancements in 3D deep learning have demonstrated the feasibility of designing specialized convolution operators for the direct processing of point cloud data within neural networks. Nevertheless, a common limitation lies in the lack of guaranteed rotation invariance, resulting in neural networks that struggle to generalize effectively when confronted with arbitrary rotations. *Zhang et al. (2019*, *2020b)* argue that convolution operators encounter greater complexity in capturing

rotation-invariant features compared to handling coordinates. Consequently, their approach involves the enhancement of convolution operators in point cloud learning by harnessing low-level geometric features inherently invariant to rotation, such as distance and angle measurements, along with incorporating global contextual information. Related research endeavors have also concentrated on refining the representation of input features and optimizing the local information modeling capabilities. *Li et al. (2021)*, for instance, has introduced a novel low-level representation that is entirely invariant to rotations, serving as a substitute for the conventional 3D Cartesian coordinates as the input to neural networks. Furthermore, *Zhang et al. (2020a)* have introduced a novel neural network called Aligned Edge Convolutional Neural Network (AECNN), explicitly for learning feature representations of point clouds with respect to Local Reference Frames (LRFs). This framework improves the ability to model the same object from different angles. RIConv++ (*Zhang, Hua & Yeung, 2022*) has proposed a convolutional operator that takes into account the relationships between points of interest and their neighboring points, as well as the intra-neighborhood relationships. This holistic approach enhances feature differentiation by constructing potent rotation-invariant features from localized regions, further advancing the capabilities of point cloud processing within neural networks.

# METHOD

We propose LSPconv, a novel point cloud feature encoder that exploits the local geometric feature of the point cloud. An LSPconv block consists of three key modules. Feature Weight Assignment module is proposed to dynamically assign weights to each neighbor point, which aims at reinforcing heterogeneity across neighborhoods by leveraging relative distance features. An asymmetrical local feature aggregator called Anisotropic Relative Feature Encoding module is designed to aggregate local features, which takes full account of the relative relationship between neighboring points and center points. Simultaneously, a local point cloud geometry encoding module called Local Spatial Projection module is proposed to obtain a better representation of the local geometric information. A detailed module design for LSPConv is illustrated in Fig. 1.

## Network architecture

We design the network architectures for point segmentation tasks and cloud classification using the proposed LSPConv layer.

**Segmentation.** We design two network architectures for part segmentation tasks and indoor segmentation using the proposed LSPConv layer. Both of them are encoder-decoder residual architecture, which is composed of a network for extracting point cloud information and a classification header for segmentation.

For part segmentation, the encoder is composed of four LSPConv layers with different point resolutions, and each residual layer is stacked by one residual block. The decoder utilizes the features from four different resolution point cloud layers and utilizes a shared-MLP for classification.

For indoor segmentation, the encoder is composed of four LSPConv layers with different point resolutions, and each residual layer is stacked with two residual blocks. The

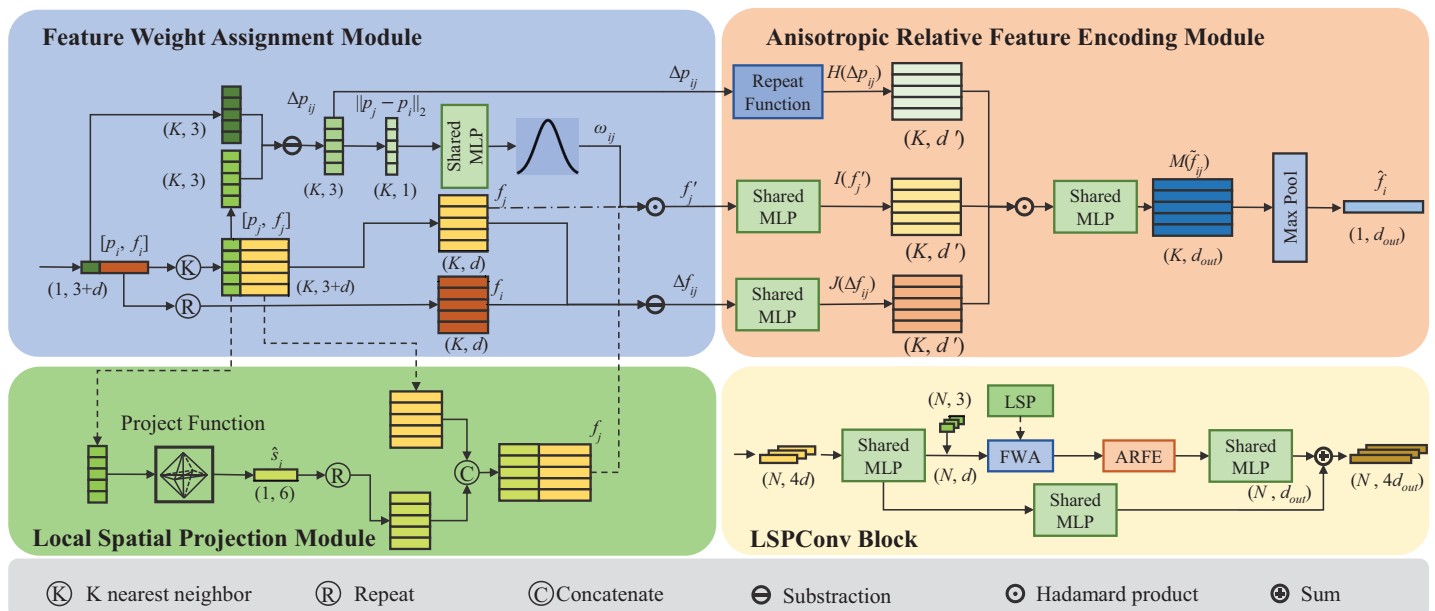

**Figure 1 The figure depicts the proposed LSPConv.** The LSPConv block is built by linking three pivotal modules together using a residual structure, as illustrated in the subfigure at the bottom right corner. The detailed composition of these three key modules is depicted in the remaining three subfigures. In the first layer, the Local Spatial Project Module will be enabled, and the tensor will be processed along the dashed line. In the subsequent layer, the Local Spatial Project Module will remain inactive, and the tensor will be processed along the dotted line.

decoder is composed of four up-sampling layers with different resolutions and an embedding layer as a classification header. Each-up sampling layer integrates the semantic information from the residual blocks in the encoder with the same resolution, ensuring the effective transmission of fine-grained semantic information.

**Classifacation.** The classification network has the same encoder component as the segmentation model. We employ dynamic graph structures instead of downsampling and interpolation for sparser point clouds in the ModelNet40 and ScanObjectNN classification datasets. Specifically, rather than being fixed using geographical locations, the network structure is changed in each layer based on feature similarity across points.

The network architecture is shown in Fig. 2.

## Feature weight assignment module

In the previous process of local information aggregation, the strength of the relationship between the feature vectors of different neighboring points and the feature vectors of the central query point is not considered. To address this issue, we propose a scheme of artificially setting a weight parameter to weigh the feature vectors of different neighbor points depending on the relative distance of the pair of points. For a set of local point clouds, its center query point is defined as point $i$ and one of the neighbor points is denoted as point $j$, $j \in \mathbf{N}(i)$. The feature of the $j$-th point after weighting can be represented in the following formula:
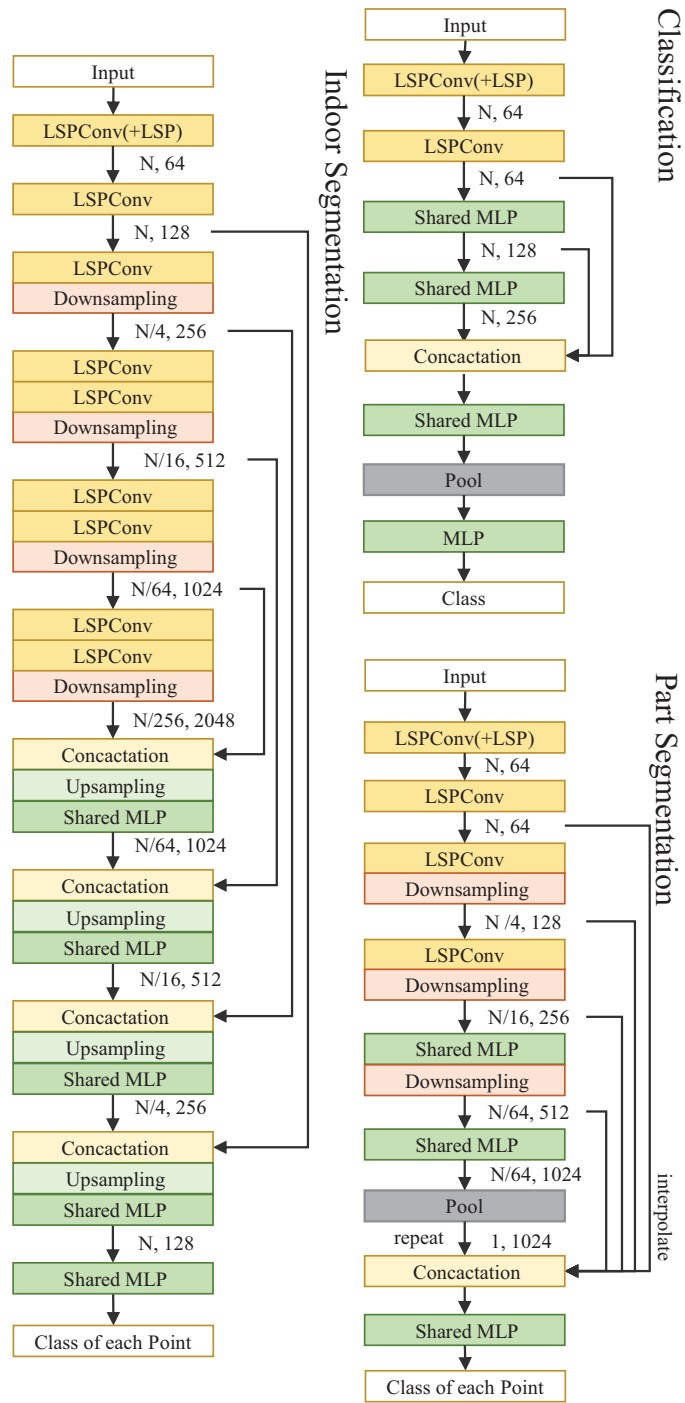

**Figure 2 Illustration of LSPConv networks for semantic segmentation and classification.** We add a local spatial projection module in the LSPConv for the first layer. Nearest neighbor interpolation is applied for upsampling.

$$f'_j = w_{ij}f_j \quad j \in \mathbf{N}(i), \tag{1}$$

where $f_j$ denotes the feature of the $j$-th neighbor point, $w_{ij}$ denotes the weight relative to the center point, and $f'_j$ denotes the output feature of it after the process of the Feature Weight

Assignment Module. Here, $w_{ij}$ is a function of the distance between the $j$-th neighbor point and the center point $i$. The corresponding $j$-th points are multiplied by $w_{ij}$ to adjust the feature of each neighbor point.

We have introduced a set of three Feature Weight Assignment modules, each designed as a function of relative distance. This innovative approach allows for the effective assignment of distinct weights to points located at varying distances from the central point. This adaptability in weight assignment facilitates enhanced modeling of the point cloud's spatial characteristics.

**Linear function distance weighting module**. The linear function distance weighting module utilizes the linear distance to weight the feature vector, which utilizes the $L_2$ norm of the relative coordinate position of each neighbor point as the weight of the feature vector of each neighbor point, as shown in the following formula:

$$w_{ij} = ||p_j - p_i||_2 \quad j \in \mathbf{N}(i), \tag{2}$$

where $p_i$ and $p_j$ denote the coordinates of the center point and its $j$-th neighbor point, $||\cdot||_2$ denotes the $L_2$ norm of the vector. With this setting, points further away from the center have greater weight, which means that the weight is positively correlated with the relative distance, forcing the central point to collect information from distant points to enrich its own features.

**Exponential function distance weighting module**. The exponential function distance weighting module utilizes the exponential distance to weight the feature vector, which utilizes the exponential function of the $L_2$ norm of the relative coordinate position of each neighbor point as the weight of the feature vector of each neighbor point, as shown in the following formula:

$$w_{ij} = e^{-||p_j - p_i||_2} \quad j \in \mathbf{N}(i). \tag{3}$$

**Radial basis function distance weighting module**. The radial basis function distance weighting module utilizes the radial basis function to weight the feature vector and utilizes the radial basis function of the relative coordinate position of each neighbor point as the weight of the feature vector of each neighbor point. The radial basis function is a kind of scalar function that is symmetric in the radial direction, which is defined as a monotone function of Euclidean distance from any point in space to a center. In our method, the radial basis function is utilized as the weighting function, as shown in the following formula:

$$w_{ij} = k(||p_j - p_i||) = e^{-\frac{||p_j - p_i||_2^2}{2\sigma^2}} \quad j \in \mathbf{N}(i), \tag{4}$$

the variance of the radial basis function is determined by the parameter $\sigma$. The larger the value $\sigma$, the wider the frequency band of the radial basis function and the better the smoothness is. By adjusting the smoothness parameter $\sigma$, a compromise can be achieved between feature over-smoothing and under-smoothing. Due to the non-uniformity of the density of the point cloud, the distribution from the neighboring points to the center point

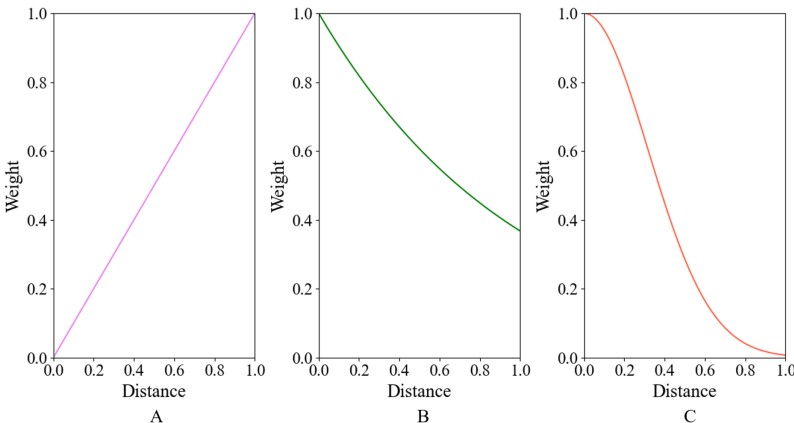

**Figure 3 Schematic diagram of weights assigned by the Feature Weight Assignment Module to neighbor points at different distances.** (A–C) denote linear function, exponential function, and radial basis distance function weighting module respectively.

is different for different center points, therefore different parameters should be selected for different center points to ensure that the Radial basis function will not produce over smooth and under smooth. To select the appropriate $\sigma$ for different center points, the learnable parameter $\sigma$ is designed according to the distribution from the neighbor point to the center point. The expression of $\sigma$ is as follows:

$$\sigma = Linear\left(max\left(||p_j - p_i||_2\right)\right) \quad j \in \mathbf{N}(i). \tag{5}$$

The design of the exponential function distance weighting module and the radial basis function distance weighting module is basically similar on a logical level, which means that the weight is negatively correlated with the relative distance. The central point is required to pay more attention to the features of its closer neighbors, for in common reality, the closer points tend to be of the same type, which emphasizes the consistency of a set of points. The weights of the above three methods as a function of distance are qualitatively depicted in Fig. 3.

Experiments show that the Feature Weight Assignment Module can significantly improve the accuracy of the model, which will be described in detail in the Evaluation section.

## Anisotropic relative feature encoding module

The point cloud local feature aggregation module has the functions of information combing and information collection, whose performance directly affects the effectiveness of the network. Therefore, the construction of an effective point cloud local feature aggregation module is the most important issue in building the whole network. In our method, the proposed Anisotropic Relative Feature Encoding Module applies both relative spatial coordinates and relative feature information to encode the feature of the neighbor point, which significantly amplifies the heterogeneity of each neighboring point, effectively

enhancing the model's capacity to capture the diversity of information among adjacent points. The general formula of the local aggregation operator is:

$$\widehat{f_i} = R\big(G\{(\Delta p_{ij}, \Delta f_{ij}, f_j) \mid j \in \mathbf{N}(i)\}\big), \tag{6}$$

where $G(\cdot)$ represents the encoding function, which is used by the local aggregation layer to encode the relative coordinates of each point $i$ and the $j$-th point into a new feature vector. To fuse all the transformed neighborhood features to form the output features of point $i$, we use $R(\cdot)$ as the reduction function, which can use the maximum, mean, or sum. $f_j$ is defined as the feature of the $j$-th point, $\Delta p_{ij}$ is defined as the coordinate of the $j$-th point relative to the $i$-th point, and $N(i)$ is defined as the neighborhood of the $i$-th point.

In order to calculate the aggregation weight of all adjacent points, a convolution filter is defined at any relative position based on the adaptive weight method, as shown in the following formula:

$$G(\Delta p_{ij}, \Delta f_{ij}, f_j) = M\big(H(\Delta p_{ij}) \odot (J(\Delta f_{ij}) \odot I(f_j))\big). \tag{7}$$

Here, $H(\cdot)$ denotes a repetition function, which is introduced to repeat the coordinate information of three dimensions several times until its dimension is approximately equal to the feature dimension. $\odot$ denotes the Hadamard product, $I(\cdot)$, $M(\cdot)$ and $J(\cdot)$ were defined as the shared-MLP.

With the Feature Weight Assignment Module, the general formula of the local aggregation operator finally becomes:

$$G(\Delta p_{ij}, \Delta f_{ij}, f_j) = M\Big(H(\Delta p_{ij}) \odot (J(\Delta f_{ij}) \odot I(f_j'))\Big). \tag{8}$$

It can be seen from Eq. (8) that $\Delta p_{ij}$ and $\Delta f_{ij}$ assign different relative feature information to different neighboring points, which distinct one certain neighbor point with others, exhibiting anisotropy. At the same time, employing only relative coordinates instead of absolute coordinates for encoding also ensures translation invariance, which is more suitable for point cloud scenes under different spatial locations.

## Local spatial projection module

In our method, a point in space and its neighbors found by the K-nearest neighbor (KNN) algorithm is considered as a local point cloud block. Existing methods merely stack the features of neighboring points together as input features. We argue that this kind of approach is insufficient because it treats the local point cloud as a set of discrete points, lacking a comprehensive description of the overall geometric characteristics of the local point cloud. In order to analyze the integrity of local point cloud blocks, we designed a point cloud local information extraction module called the Local Spatial Projection Module, which models the geometric shape features of a point cloud through the utilization of vector projection and reassembly methods.

In the construction of the point cloud shape information extraction module, the center point of the local point cloud block is first used as the origin to build up a Cartesian

coordinate system and regards the relative position of each point in the local point cloud block to the origin as the relative position vector of each point. Then, we divide the space Cartesian coordinate system into six axes: $x$ positive semi-axis, $x$ negative semi-axis, $y$ positive semi-axis, $y$ negative semi-axis, $z$ positive semi-axis, $z$ negative semi-axis. In order to obtain the complete representation of the local point cloud, the module projects the vector represented by the relative coordinate of each neighbor point to the six axes and sums the signals on each axis. The above process can be represented by the following formula:

$$s_{i+}^{l} = \sum_{j} \max\left(0, \Delta p_{ij}^{l}\right), \quad j \in \mathbf{N}(i), l = x, y, z, \tag{9}$$

$$s_{i-}^{l} = \sum_{j} \max\left(0, -\Delta p_{ij}^{l}\right), \quad j \in \mathbf{N}(i), l = x, y, z, \tag{10}$$

where $p_i$ and $p_j$ denote the coordinates of the center point and its $j$-th neighbor point, $\Delta p_{ij}^{l}$ denotes the components of the relative coordinate vector on the $l$-axis, $l = x, y, z$, and $s_{i+}^{l}/s_{i-}^{l}$ denote the signal obtained by the summation operator on the $l$ positive/negative semi-axis.

After the projection operation, the original space shape is obtained. To reduce the sensitivity of the network to the change in point cloud density, the original space shape composition of signals projected on the six axes is normalized.

We proposed three normalization methods:

$L_2$ **normalization** The first method utilized $L_2$ normalization to normalize the original space shape, as shown below:

$$s_{i*}^{l} = \frac{s_{i*}^{l}}{\sqrt{\sum_{m} s_{i+}^{m\,2} + \sum_{m} s_{i-}^{m2}}} \quad m = x, y, z \quad * = +/-, \tag{11}$$

where $s_{i*}^{l}$ denotes the signal obtained by the summation operator on the $l$ positive/negative semi-axis and $s_{i*}^{l}$ is defined as the normalized signal on the $l$ positive/negative semi-axis.

**Adaptive normalization.** The second method utilized a multilayer perceptron to adaptively adjust the length of each axis of the original space shape to achieve a better spatial representation, as shown below:

$$s_i = LeakReLu(BN(Linear(s_i))), \tag{12}$$
$$s_i = ReLu(BN(Linear(s_i))), \tag{13}$$

$s_i$ is the initial input, $s_i$ is the middle represents and $s_i$ is the final result obtained. Here we employ the *ReLu* activation function to ensure that the output signal of each axis is non-negative.

**Sharp normalization.** The third method is called sharp normalization. This method normalizes all the non-zero signals to 1 and remains the zero signal, which can be easily represented by the following formula:

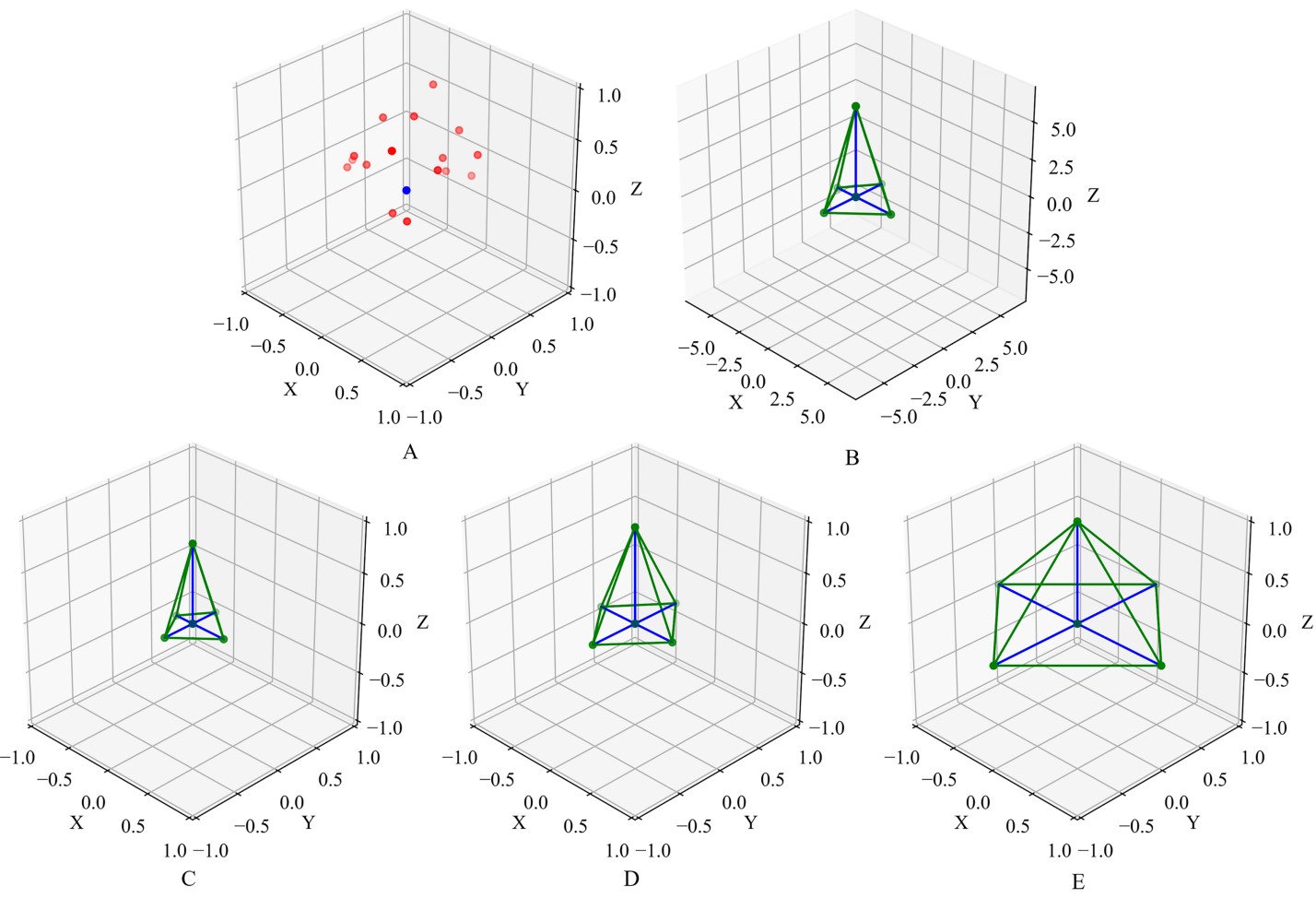

**Figure 4 The effect of Local Spatial Projection module.** (A) Demonstrate a set of local points. The blue point indicates the center query point, and the red points indicate its neighbor points found by the KNN algorithm. (B) Is the original space shape attained by the neighbor points projection and (C–E), demonstrate the space shapes obtained by $L_2$ normalization, adaptive normalization and sharp normalization methods.

$$s_{i*}^l = \begin{cases} 0, & s_{i*}^l = 0 \\ 1, & s_{i*}^l > 0 \end{cases}. \tag{14}$$

By this design, the network is principally devoted to the distinction between points on the boundary and points in the interior. The sharp normalization method is proven to be efficient for indoor segmentation.

Different normalization methods are qualitatively depicted in Fig. 4. After the normalization, the normalized space shape, the relative position coordinate information, and color information are input into the neural network as the basic information of the point cloud for training. Note Local Spatial Projection Module exclusively captures the local shape of the original point cloud and is not applied to the downsampled point cloud. This design is because, while downsampling aggregates neighboring features and expands

the receptive field, it also disrupts fine-grained geometric structures and appearances. Therefore, it is advisable to use this module exclusively at the highest resolution. In the Evaluation section, it is proved that adding space shape can effectively improve the network's ability to capture the local information of the point cloud compared with simply inputting the relative position coordinate information.

## EVALUATION

In this section, we evaluate our model on four different datasets to achieve point cloud part segmentation, indoor segmentation, and classification tasks including ShapeNetPart, Stanford 3D Indoor Space (S3DIS) ModelNet40, and ScanObjectNN datasets. Comprehensive network configuration and comparisons are available for each task.

### Part segmentation

**Data**. In order to improve our model, we do more things. On the ShapeNetPart dataset (*Yi et al., 2016*), we further test our model for the part segmentation task. A total of 16,881 shapes totaling 16 categories make up this dataset, with 14,006 used for training and 2,874 for testing. Each point has a label assigned from a pool of 50 parts, and each point cloud has between two and six parts. We replicate *Qi et al. (2017b)*'s experimental setup and use their supplied data for benchmarking. Each form has 2,048 points sampled from it. In addition to the 3D coordinates, the input attributes also include the point normal.

**Implementation**. We select $L_2$ normalization for Local Spatial Projection Module to normalize the space shape and select radial basis function distance weighting module for Feature Weight Assignment Module to assign weights to different neighbor points for part segmentation task. Based on Pointnet (*Qi et al., 2017a*), a one-hot vector representing the different categories for each point is used to compute the segmentation results. Other training parameters are the same as for classification. All layers take 20 for the number $k$ of neighborhood size. To obtain the global feature, the max-pooling function is chosen. We select LeakyReLU for the activation function with a negative slope setting to 0.1 and batch normalization. We use SGD optimizer with momentum setting to 0.9 (*Wang et al., 2019b*). We apply a cosine annealing schedule to adjust the learning rate during the training. The initial learning rate is 0.1 and is decreased until 0.001 (*Loshchilov & Hutter, 2016*). The batch size for the training models is 32. We use PyTorch implementation and train the network on a Tesla A40 GPU.

**Results.** In Table 1, we present the mean instance IoU (mIoU) and mean class IoU (mcIoU). The IoU of a shape is calculated by averaging the IoU of each part, in accordance with the evaluation scheme of *Qi et al. (2016)*. By averaging the IoUs overall testing instances, the mean IoU (mIoU) is calculated. The mean IoU across all form categories is known as the class IoU (mcIoU). The mean IoU of LSPconv on ShapeNetPart dataset reaches 86.6%, surpassing the previous competitive methods PointMLP (*Ma et al., 2022*) and Kpconv (*Thomas et al., 2019*) 0.2% and 0.3% respectively. We also exhibit the results of class-based segmentation. Comparing LSPconv to other approaches, it performs at the cutting edge.

**Table 1 Part segmentation results on ShapeNet dataset evaluated as the mean class IoU (mcIoU) and mean instance IoU (mIoU).** The bold text indicates the best result.

| Method | mcIoU | mIoU | Air plane | Bag | Cap | Car | Chair | Ear phone | Guitar | Knife | Lamp | Laptop | Motor bike | Mug | Pistol | Rocket | Skate board | Table |
|---|---|---|---|---|---|---|---|---|---|---|---|---|---|---|---|---|---|---|
| PointNet (*Qi et al., 2017a*) | 80.4 | 83.7 | 83.4 | 78.7 | 82.5 | 74.9 | 89.6 | 73.0 | 91.5 | 85.9 | 80.8 | 95.3 | 65.2 | 93.0 | 81.2 | 57.9 | 72.8 | 80.6 |
| PointNet++ (*Qi et al., 2017b*) | 81.9 | 85.1 | 82.4 | 79.0 | 87.7 | 77.3 | 90.8 | 71.8 | 91.0 | 85.9 | 83.7 | 95.3 | 71.6 | 94.1 | 81.3 | 58.7 | 76.4 | 82.6 |
| DGCNN (*Wang et al., 2019b*) | 82.3 | 85.2 | 84.0 | 83.4 | 86.7 | 77.8 | 90.6 | 74.7 | 91.2 | 87.5 | 82.8 | 95.7 | 66.3 | 94.9 | 81.1 | 63.5 | 74.5 | 82.6 |
| PointASNL (*Yan et al., 2020*) | – | 86.1 | 84.1 | 84.7 | 87.9 | 79.7 | 92.2 | 73.7 | 91.0 | 87.2 | 84.2 | 95.8 | 74.4 | 95.2 | 81.0 | 63.0 | 76.3 | 83.2 |
| 3D-GCN (*Lin, Huang & Wang, 2021*) | 82.1 | 85.1 | 83.1 | 84.0 | 86.6 | 77.5 | 90.3 | 74.1 | 90.9 | 86.4 | 83.8 | 95.6 | 66.8 | 94.8 | 81.3 | 59.6 | 75.7 | 82.8 |
| KPConv (*Thomas et al., 2019*) | 85.1 | 86.4 | 84.6 | 86.3 | 87.2 | 81.1 | 91.1 | 77.8 | 92.6 | 88.4 | 82.7 | 96.2 | 78.1 | 95.8 | 85.4 | 69.0 | 82.0 | 83.6 |
| AGConv (*Zhou et al., 2021*) | 83.4 | 86.4 | 84.8 | 81.2 | 85.7 | 79.7 | 91.2 | 80.9 | 91.9 | 88.6 | 84.8 | 96.2 | 70.7 | 94.9 | 82.3 | 61.0 | 75.9 | 84.2 |
| PAConv (*Xu et al., 2021*) | 84.2 | 86.0 | – | – | – | – | – | – | – | – | – | – | – | – | – | – | – | – |
| Point Trans. (*Zhao et al., 2021*) | 83.7 | 86.6 | – | – | – | – | – | – | – | – | – | – | – | – | – | – | – | – |
| PointMLP (*Ma et al., 2022*) | 84.6 | 86.1 | 83.5 | 83.4 | 87.5 | 80.5 | 90.3 | 78.2 | 92.2 | 88.1 | 82.6 | 96.2 | 77.5 | 95.8 | 85.4 | 64.6 | 83.3 | 84.3 |
| PointNeXt (*Qian et al., 2022*) | 84.4 | 86.7 | – | – | – | – | – | – | – | – | – | – | – | – | – | – | – | – |
| StratifiedTransformer (*Lai et al., 2022*) | 85.1 | 86.6 | – | – | – | – | – | – | – | – | – | – | – | – | – | – | – | – |
| SPoTr (*Park et al., 2023*) | **85.4** | **87.2** | – | – | – | – | – | – | – | – | – | – | – | – | – | – | – | – |
| APES (*Wu et al., 2023*) | 83.7 | 85.8 | – | – | – | – | – | – | – | – | – | – | – | – | – | – | – | – |
| Ours | 84.2 | 86.6 | 84.9 | 84.4 | 88.8 | 81.6 | 91.9 | 76.6 | 91.8 | 87.5 | 85.9 | 96.4 | 76.4 | 95.4 | 82.5 | 64.5 | 76.2 | 83.2 |

## Indoor segmentation

**Data**. We evaluate our model on Stanford Large-Scale 3D Indoor Spaces Dataset (*Armeni et al., 2016*) (S3DIS). This dataset consists of 271 rooms in total. Each room includes six 3D scan point clouds for indoor areas, belonging to 13 semantic categories. For a common evaluation protocol (*Tchapmi et al., 2017*; *Qi et al., 2017a*; *Thomas et al., 2019*), we select Area 5 to test our model because it is not in the same building as other areas.

**Implementation**. We select Sharp normalization for Local Spatial Projection Module to normalize the space shape and select radial basis function distance weighting module for Feature Weight Assignment Module to assign weights to different neighbor points for indoor segmentation task. The massive indoor scene datasets lead to further challenges, including the larger scale of the scene in a real environment with a lot of further noise and distinct outlines. Therefore, we follow the experimental settings of KPConv (*Thomas et al., 2019*), and train our network by applying randomly sample clouds in spheres. The subclouds consist of more points with various sizes and are stacked into batches for training our network much further. During the test phase, spheres are equably picked in the scenes. To ensure the accuracy of the test, we ensure that every point is examined several times by using a voting scheme. The input point attributes contain the RGB colors and the original heights. We use SGD optimizer with momentum set to 0.98 and an initial learning rate of 0.01. The learning rate decays by multiplying $0.1^{0.02}$ in each epoch. To prevent gradient explosion, the gradient clipping threshold value is set to 100. We use PyTorch implementation and train the network on a Tesla A40 GPU. Differing from the part segmentation setting, we omit the $J(\Delta f_{ij})$ in Anisotropic Relative Feature Encoding Module for such a setting that attains better performance in the experiment.

**Table 2 Semantic segmentation results on S3DIS dataset evaluated on Area 5.** The bold text indicates the best result.

| Method | OA | mAcc | mIoU | Ceiling | Floor | Wall | Beam | Column | Window | Door | Table | Chair | Sofa | Bookcase | Board | Clutter |
|---|---|---|---|---|---|---|---|---|---|---|---|---|---|---|---|---|
| PointNet (Qi et al., 2017a) | – | 49.0 | 41.1 | 88.8 | 97.3 | 69.8 | 0.1 | 3.9 | 46.3 | 10.8 | 59.0 | 52.6 | 5.9 | 40.3 | 26.4 | 33.2 |
| SegCloud (Tchapmi et al., 2017) | – | 57.4 | 48.9 | 90.1 | 96.1 | 69.9 | 0.0 | 18.4 | 38.4 | 23.1 | 70.4 | 75.9 | 40.9 | 58.4 | 13.0 | 41.6 |
| PointASNL (Yan et al., 2020) | 87.7 | 68.5 | 62.6 | 94.3 | 98.4 | 79.1 | 0.0 | 26.7 | 55.2 | 66.2 | 83.3 | 86.8 | 47.6 | 68.3 | 56.4 | 52.1 |
| PointCNN (Li et al., 2018b) | 85.9 | 63.9 | 57.3 | 92.3 | 98.2 | 79.4 | 0.0 | 17.6 | 22.8 | 62.1 | 74.4 | 80.6 | 31.7 | 66.7 | 62.1 | 56.7 |
| PointWeb (Zhao et al., 2019) | 87.0 | 66.6 | 60.3 | 92.0 | 98.5 | 79.4 | 0.0 | 21.1 | 59.7 | 34.8 | 76.3 | 88.3 | 46.9 | 69.3 | 64.9 | 52.5 |
| KPConv (Thomas et al., 2019) | – | 72.8 | 67.1 | 92.8 | 97.3 | 82.4 | 0.0 | 23.9 | 58.0 | 69.0 | 81.5 | 91.0 | 75.4 | 75.3 | 66.7 | 58.9 |
| PosPool (Liu et al., 2020) | – | – | 66.7 | – | – | – | – | – | – | – | – | – | – | – | – | – |
| PAConv (Xu et al., 2021) | – | 73.0 | 66.6 | 94.6 | 98.6 | 82.4 | 0.0 | 26.4 | 58.0 | 60.0 | 89.7 | 80.4 | 74.3 | 69.8 | 73.5 | 57.7 |
| BAAF-Net (Qiu, Anwar & Barnes, 2021c) | 88.9 | 73.1 | 65.4 | – | – | – | – | – | – | – | – | – | – | – | – | – |
| AGConv (Zhou et al., 2021) | 90.0 | 73.2 | 67.9 | 93.9 | 98.4 | 82.2 | 0.0 | 23.9 | 59.1 | 71.3 | 91.5 | 81.2 | 75.5 | 74.9 | 72.1 | 58.6 |
| PointNeXt-L (Qian et al., 2022) | 90.0 | – | 69.0 | – | – | – | – | – | – | – | – | – | – | – | – | – |
| StratifiedTransformer (Lai et al., 2022) | **91.5** | **78.1** | **72.0** | – | – | – | – | – | – | – | – | – | – | – | – | – |
| SPoTr (Park et al., 2023) | 90.7 | 76.4 | 70.8 | – | – | – | – | – | – | – | – | – | – | – | – | – |
| SPT (Robert, Raguet & Landrieu, 2023) | – | – | 68.9 | – | – | – | – | – | – | – | – | – | – | – | – | – |
| Ours | 90.2 | 74.0 | 68.2 | 94.4 | 98.4 | 83.0 | 0.0 | 26.0 | 60.2 | 71.4 | 90.4 | 81.1 | 75.5 | 70.0 | 74.9 | 61.6 |

**Results.** In Table 2, we present the average classwise intersection over union (mIoU), average classwise accuracy (mAcc), and average classwise accuracy (OA). Each class's IoU result is also supplied. LSPconv reaches 90.2%, 74.0%, and 68.2% under OA, mAcc, and mIoU metrics respectively. The performance exceeds that of previous classic models such as KPConv (*Thomas et al., 2019*), PAconv (*Xu et al., 2021*) and AGConv (*Zhou et al., 2021*). The remarkable performance on difficult large scene semantic segmentation datasets further demonstrates the effectiveness of the proposed local spatial projection module, anisotropic kernel, and weight assignment module.

## Point cloud classification

**Data.** We test our model using two datasets: ModelNet40 (*Wu et al., 2015*) and ScanObjectNN (*Uy et al., 2019*) to Validate the classification ability of LSPConv.

ModelNet40 is subdivided into 40 categories and consists of 12,311 meshed CAD models. A total of 9,843 CAD models are intended to train our model, while 2,468 CAD models are intended to test it. We sample 2,024 points uniformly for each object and use the $(x, y, z)$ coordinates of them as input.

While ModelNet40 has long been considered the standard benchmark for point cloud analysis, the rapid evolution of point cloud analysis techniques may render it inadequate for assessing the capabilities of modern methods. In light of this, we have also undertaken experiments using the ScanObjectNN benchmark. ScanObjectNN comprises approximately 15,000 real-world scanned objects, meticulously classified into 15 distinct categories, with a total of 2,902 unique object instances. The presence of occlusions and noise in ScanObjectNN presents formidable challenges to existing point cloud analysis

**Table 3 Classification results on Modelnet40 dataset.** The bold text indicates the best result.

| Method | mAcc (%) | OA (%) |
|---|---|---|
| VoxNet (*Maturana & Scherer, 2015*) | 83.0 | 85.9 |
| Subvolume (*Qi et al., 2016*) | 86.0 | 89.2 |
| PointNet (*Qi et al., 2017a*) | 86.0 | 89.2 |
| PointNet++ (*Qi et al., 2017b*) | – | 91.9 |
| Kd-Net (*Klokov & Lempitsky, 2017*) | – | 90.6 |
| SpidcrCNN (*Xu et al., 2018*) | – | 92.4 |
| PointCNN (*Li et al., 2018b*) | 88.1 | 92.2 |
| SO-Net (*Li, Chen & Lee, 2018*) | – | 93.4 |
| DGCNN (*Wang et al., 2019b*) | 90.2 | 92.9 |
| KPConv (*Thomas et al., 2019*) | – | 92.9 |
| 3D-GCN (*Lin, Huang & Wang, 2020*) | – | 92.1 |
| PointASNL (*Yan et al., 2020*) | – | 93.2 |
| Point Trans. (*Zhao et al., 2021*) | 90.6 | 93.7 |
| PointMLP (*Ma et al., 2022*) | 91.4 | 94.5 |
| PointNeXt (*Qian et al., 2022*) | 90.8 | 93.2 |
| Point2Vec (*Abou Zeid et al., 2023*) | – | **94.8** |
| APES (*Wu et al., 2023*) | – | 93.8 |
| DeLA (*Chen et al., 2023a*) | **92.2** | 94.0 |
| Ours | 90.5 | 93.2 |

approaches. For our experiments, we have chosen to tackle the most challenging perturbed variant, denoted as PB_T50_RS.

**Implementation**. We select $L_2$ normalization for the Local Spatial Projection Module to normalize the space shape and select the radial basis function distance weighting module for the Feature Weight Assignment Module to assign weights to different neighbor points for the point cloud classification task. We recompute the graph in accordance with the network configurations (*Wang et al., 2019b*) based on the similarity of the feature sets in each layer. For all layers, the $k$ of neighborhood size is set to 20. The multiscale features are aggregated using shortcut connections and fed forward to a shared fully connected layer. Max-pooling is used to acquire the global feature. LeakyReLU and batch normalization are used on all levels. We utilize AdamW optimizer and apply a cosine annealing schedule to adjust the learning rate during the training. The initial learning rate is 0.001 and is decreased until 0.0001 (*Loshchilov & Hutter, 2017*). Each training model has a batch size of 32. We train the network on a Tesla A40 GPU using the PyTorch implementation. The data augmentation procedure includes point shifting, scaling, and perturbation.

**Results** The mean class accuracy (mAcc) and total accuracy (OA) are the evaluation metrics for both datasets. In Tables 3 and 4, we display the classification results on the ModelNet40 dataset and the ScanObjectNN dataset respectively. For the ModelNet40 dataset, we compare the input data types and the number of points connected to each method. With a limited input size of 2 k points, our model obtains a desirable outcome on

**Table 4** **Classification results on ScanObjectNN (PB_T50_RS) dataset.** The bold text indicates the best result.

| Method | mAcc (%) | OA (%) |
|---|---|---|
| PointNet (*Qi et al., 2017a*) | 63.4 | 68.2 |
| SpiderCNN (*Xu et al., 2018*) | 69.8 | 73.7 |
| PointNet++ (*Qi et al., 2017b*) | 75.4 | 77.9 |
| DGCNN (*Wang et al., 2019b*) | 73.6 | 78.1 |
| PointCNN (*Li et al., 2018b*) | 75.1 | 78.5 |
| BGA-DGCNN (*Uy et al., 2019*) | 75.7 | 79.7 |
| BGA-PN++ (*Uy et al., 2019*) | 77.5 | 80.2 |
| DRNet (*Qiu, Anwar & Barnes, 2021a*) | 78 | 80.3 |
| GBNet (*Qiu, Anwar & Barnes, 2021b*) | 77.8 | 80.5 |
| SimpleView (*Goyal et al., 2021*) | – | 80.5 $\pm$ 0.3 |
| PRANet (*Cheng et al., 2021*) | 79.1 | 82.1 |
| MVTN (*Hamdi, Giancola & Ghanem, 2021*) | – | 82.8 |
| PointMLP (*Ma et al., 2022*) | 83.9 $\pm$ 0.5 | 85.4 $\pm$ 0.3 |
| PointNeXt (*Qian et al., 2022*) | 85.8 $\pm$ 0.6 | 87.7 $\pm$ 0.4 |
| Point2Vec (*Abou Zeid et al., 2023*) | – | 87.5 |
| SPoTr (*Park et al., 2023*) | **86.8** | **88.6** |
| Ours | 76.7 | 80.0 |

**Table 5** **The comparison of parameters number and overall accuracy of previous methods and ours.** The bold text indicates the best result.

| Method | #Parameters | OA (%) |
|---|---|---|
| PointNet (*Qi et al., 2017a*) | 0.6 M | 89.2 |
| PointNet++ (*Qi et al., 2017b*) | 1.5 M | 91.9 |
| DGCNN (*Wang et al., 2019b*) | 1.8 M | 92.9 |
| KPConv (*Thomas et al., 2019*) | 14.3 M | 92.9 |
| PointMLP (*Ma et al., 2022*) | 13.2 M | 94.5 |
| PointNeXt (*Qian et al., 2022*) | 1.4 M | 93.2 |
| Ours | 1.8 M | 93.2 |

the ModelNet40 dataset. The accuracy achieves 90.5% and 93.2% under mAcc and OA metrics respectively, almost on par with the advanced PointNeXt algorithm. For the ScanObjectNN dataset, accuracy has reached 76.7% and 80.0% in terms of mAcc and OA metrics respectively under a lightweight and simple model design. The experimental results conducted on these two datasets indicate that our algorithm has a satisfactory performance in classification tasks and exhibits a competitive edge.

## Efficiency

We provide the parameter values and corresponding network performances in Table 5 to contrast the complexity of our model with previous influential methods. These models are for the classification task on ModelNet40. LSPconv has very few additional learnable

**Table 6 Ablation study on ShapeNet dataset.** The bold text indicates the best result.

|  | FWA | ARFE | LSP | mcIoU (%) | mIoU (%) |
|---|---|---|---|---|---|
| EXP1 |  |  |  | 81.8 | 85.2 |
| EXP2 | ✓ |  |  | 83.2 | 85.7 |
| EXP3 | ✓ | ✓ |  | 83.8 | 86.4 |
| EXP4 | ✓ | ✓ | ✓ | **84.2** | **86.6** |

Note:
FWA, feature weight assignment module; ARFE, anisotropic relative feature encoding module; LSP, local spatial projection module.

parameters because the $L_2$ normalization in Local Spatial Projection Module does not introduce learnable parameters and the radial basis function Feature Weight Assignment Module just requires training an MLP to attain $\sigma$. From the table, it is evident that our model achieves a commendable performance, delivering an overall accuracy of 93.2% while utilizing only 1.8M parameters. This balance between accuracy and efficiency is comparable to leading-edge algorithms.

# ABLATION STUDIES

We describe some architectural choices in our network and compare LSPConv to several other ablation networks. Ablation studies are conducted on the ShapeNet dataset.

## The effect of each proposed module

To verify the specific contribution of each module, we tested each module item by item. The results are summarized in Table 6. When introducing Feature Weight Assignment Module, $w_{ij}$ will be applied to $f_j$, $f_j'$ will be utilized instead of the original $f_j$. When introducing Anisotropic Relative Feature Encoding Module, $H(\Delta p_{ij})$ and $J(\Delta f_{ij})$ will be applied Hadamard product to $I(f_j')$. When introducing Local Spatial Projection Module, the $L_2$ normalized space shape will be modeled for the first layer instead of just utilizing $\Delta p_{ij}$ for local geometric information description. The first line exhibits the experimental results of applying a symmetric kernel without introducing any of the proposed modules. The second row shows that equipped with the proposed Feature Weight Assignment (FWA) Module, the performance of the network has been boosted from 81.8% and 85.2% to 83.2% and 85.7% under mcIoU and mIoU on ShapeNet dataset. The third row demonstrates that including the proposed Anisotropic Relative Feature Encoding (ARFE) Module into the network improves the result to 83.8% and 86.4% in terms of mcIoU and mIoU, demonstrating its effectiveness of anisotropy kernel for enhancing the performance of the network. Finally, as shown in the fourth row, when applying Local Spatial Projection Module (LSP) for building a better representation of local spatial information, the results are improved to 84.2% and 86.6% in terms of mcIoU and mIoU.

## Feature weight assignment module

We discuss the different designs of the Feature Weight Assignment Modules. Three different Feature Weight Assignment Modules are designed to assign weights to each neighbor point. We tested the performance of these modules separately. The introduction

**Table 7 Results of part segmentation network with different Feature Weight Assignment modules.** 'Without' means omitting the Feature Weight Assignment module. The bold text indicates the best result.

| Weighting module | mcIoU (%) | mIoU (%) |
|---|---|---|
| Without | 83.7 | 86.4 |
| Linear function | 83.7 | 86.2 |
| Exponential function | 83.9 | 86.4 |
| Radial basis function | **84.2** | **86.6** |

**Table 8 Results of part segmentation network with different relative information input for anisotropic relative feature encoding.** The bold text indicates the best result.

| Relative information | mcIoU (%) | mIoU (%) |
|---|---|---|
| $J(\Delta f_{ij}) \odot I(f_j')$ | 82.6 | 85.6 |
| $H(\Delta p_{ij}) \odot I(f_j')$ | 84.0 | 86.3 |
| $H(\Delta p_{ij}) \odot J(\Delta f_{ij}) \odot I(f_j')$ | **84.2** | **86.6** |

of exponential function can improve the segmentation accuracy by 0.2% under the mcIoU index, and the introduction of radial basis function can improve the segmentation accuracy by 0.5% and 0.2% under the mcIoU and mIoU indexes, respectively. This proves that the strategy of assigning different weights to the features of different neighbor points according to their distance is effective. Generally, the weight is negatively correlated with the distance. The learnable parameter $\sigma$ set by the radial basis function adaptively adjusts the smoothness of the weight distribution function, which can better find better weights for different neighbor points. As shown in Table 7, LSPConv can efficiently assign weights and obtain the best performance when the radial basis function distance weighting module is applied.

### Relative information selection

In LSPConv, we imply relative information $\Delta p_{ij}$ and $\Delta f_{ij}$ for local feature modeling. As shown in Table 8, $\Delta p_{ij}$ and $\Delta f_{ij}$ are both utilized in order to enhance the difference between different neighbor points, which means the kernel could show anisotropy to model the local feature of the point cloud. We explore the performance of different inputs of relative information. When only $\Delta f_{ij}$ is introduced, the accuracy is achieved 82.6% and 85.6% under mcIoU and mIoU indexes, respectively. Similarly, when only $\Delta p_{ij}$ is introduced, the accuracy is achieved 84.0% and 86.3% under the mcIoU and mIoU indexes respectively. The experimental results exhibit that compared with utilizing $J(\Delta f_{ij})$ and $H(\Delta f_{ij})$ alone, simultaneously applying two relative information, $\Delta p_{ij}$ and $\Delta f_{ij}$, is the best way to encode the features of each point, with the mcIoU and mIoU indexes achieving 84.2% and 86.6% respectively.

**Table 9 Results of part segmentation network with different local shape information representation.** '$\Delta p_{ij}$ only' means replacing local spatial projection information with relative coordinates. The bold text indicates the best result.

| Local shape information | mcIoU (%) | mIoU (%) |
|---|---|---|
| $\Delta p_{ij}$ only | 83.7 | 86.2 |
| Adaptive normalization | 84.0 | 86.4 |
| $L_2$ normalization | **84.2** | **86.6** |
| Sharp normalization | 83.9 | 86.3 |

**Table 10 Results of part segmentation network on ShapNet with different numbers $k$ of nearest neighbors.** The bold text indicates the best result.

| Number $k$ of neighbors | mcIoU (%) | mIoU (%) |
|---|---|---|
| 5 | 81.6 | 85.2 |
| 10 | 83.3 | 86.1 |
| 20 | **84.2** | **86.6** |
| 40 | 84.0 | 86.4 |

Notice that the module exhibits the strongest anisotropy when all relative features are utilized and consequently the best results are obtained.

### Local shape information representation

Furthermore, we discuss the different designs of the local shape information representation. There are three different Local Spatial Projection modules designed to model the local shape of the point cloud. We tested the performance of these three modules separately, as shown in Table 9. Experiments were carried out on different Local Spatial Projection modules. The adaptive normalization based Local Spatial Projection module is trained to adaptively scale the original space shape, compared to just using $\Delta p_{ij}$, The performance of part segmentation is improved by 0.3% and 0.2% under mcIoU and mIoU indexes. $L_2$ regularization of original space shape based on the local spatial projection module of $L_2$ normalization, the performance of part segmentation is improved by 0.5% and 0.4% under mcIoU and mIoU indexes. The Sharp normalization method is more appropriate for scene semantic segmentation, but embedding it in part segmentation tasks still works, and the performance of part segmentation has improved by 0.2% and 0.1% under the mcIoU and mIoU metrics. The test results show that the proposed Local shape information representation module can better represent the local spatial information of the point cloud compared with simple $\Delta p_{ij}$. The experimental results based on ShapeNet dataset show that all three proposed local spatial projection modules are effective, among which the $L_2$ normalization is the most effective. Notice that the $L_2$ normalization method performs better than the sharp normalization method on ShapeNet dataset, while The experiment shows the opposite results on S3DIS dataset. To this end, the $L_2$ normalization method is applied for part segmentation and classification while the sharp normalization method is selected for indoor segmentation.

### Number of neighbor points

We conduct experiments to find out the influence of the number of nearest neighbors, as shown in Table 10. The result shows that the model achieving 84.2% and 86.6% under mcIoU and mIoU indexes, performs best when the number of neighbor points $k = 20$. When the number of neighbors is too small, such as $k = 5$ or $k = 10$, the receptive field is insufficient and the local semantic information cannot be fully mined. We find that when $k$ is too large ($k = 40$), the performance of the model deteriorates to 84.0% and 86.4% under mcIoU and mIoU indexes. It is mainly because a large receptive field introduces noise from other parts, resulting in ambiguous local information and a lack of consistency in the point cloud. This result confirms that appropriately increasing the number of neighbor points can expand the receptive field, which improves the ability of the operator to capture the geometric structure and represent local information.

## CONCLUSION

In this article, we present a network for point cloud semantic segmentation, named LSPConv. LSPConv constructs a Local Spatial Projection Module to model local geometry information, which has been proven to be effective for irregular and disordered point clouds. Meanwhile, we designed a radial basis function distance feature weight assignment module to assign weights for the different points according to relative distance. Additionally, an adaptive operator called the Anisotropic Relative Feature Encoding module was introduced, which is able to encode points adaptively according to the relative feature, which forces the module to exhibit anisotropy on the basis of satisfying translation invariance. The proposed approach aims to represent point clouds more accurately and improve the local modeling capability, which can better handle irregular and disordered point clouds. Our approaches achieve impressive results for classifying and segmenting point clouds in several benchmark datasets based on the extensive qualitative and quantitative evaluation.

### Funding

This work was supported by the National Key Research and Development Program under Grant 2022YFB4702402, by the National Natural Science Foundation of China under Grant 62176072 and by the Open Research Fund of Anhui Province Key Laboratory of Machine Vision Inspection (KLMVI-2023-HIT-12). The funders had no role in study design, data collection and analysis, decision to publish, or preparation of the manuscript.

### Grant Disclosures

The following grant information was disclosed by the authors:
National Key Research and Development Program: 2022YFB4702402.
National Natural Science Foundation of China: 62176072.
Anhui Province Key Laboratory of Machine Vision Inspection: KLMVI-2023-HIT-12.

## Competing Interests

Xianshui Tao is employed by Wuhu Hit Robot Technology Research Institute Co., Ltd. The other authors declare that they have no competing interests.

## Author Contributions

- Haoming Zhang conceived and designed the experiments, performed the experiments, analyzed the data, performed the computation work, prepared figures and/or tables, authored or reviewed drafts of the article, and approved the final draft.
- Ke Wang conceived and designed the experiments, authored or reviewed drafts of the article, and approved the final draft.
- Chen Zhong conceived and designed the experiments, authored or reviewed drafts of the article, and approved the final draft.
- Kaijie Yun analyzed the data, prepared figures and/or tables, and approved the final draft.
- Zilong Wang analyzed the data, prepared figures and/or tables, and approved the final draft.
- Yifan Yang analyzed the data, prepared figures and/or tables, and approved the final draft.
- Xianshui Tao conceived and designed the experiments, authored or reviewed drafts of the article, and approved the final draft.

## Data Availability

The code and experimental results are available in the Supplemental Files.

The public dataset for classification called ModelNet40 is available at: https://3dshapenets.cs.princeton.edu/.

The public dataset for part segmentation called ShapeNetPart is available at: https://shapenet.org/.

The public dataset for indoor segmentation called S3DIS is available upon request at: https://goo.gl/forms/4SoGp4KtH1jfRqEj2.

## Supplemental Information

Supplemental information for this article can be found online at http://dx.doi.org/10.7717/peerj-cs.1738#supplemental-information.

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
