# Peer review of "LSPConv: local spatial projection convolution for point cloud analysis"

_PeerJ Computer Science, doi:10.7717/peerj-cs.1738_

## Round 0.1 · original submission · Major Revisions

According to the reviewer's opinion, the manuscript still needs major revision. The author needs to further revise the manuscript according to the comments, especially the experimental part.

Please ensure your work is clearly discussed in the context of contemporary literature.

Reviewer 1 ·

Basic reporting

this paper proposes a Local Spatial Projection Module to improve a network for point cloud classification and semantic segmentation. this paper is acceptable with some modification.
1. In Introduction, the author said that “analyzing 3D information from point clouds using deep learning techniques is more challenging than that from 2D images due to its sparse, irregular, and unordered structure”, “This type of processing results in spatial information lost or repeatedly represented”. Please add relevant references to assist in explanation.
2. Based on logical considerations, it is recommended to adjust the order of some chapters, the section “ABLATION STUDIES” should be before section “EVALUATION”.

Experimental design

In line 227-233, The author has provided some preset parameters required in the network, such as “momentum setting to 0.9”. Please add references or demonstrate the rationality of these parameter settings through experimental analysis.

Validity of the findings

The author has conducted sufficient experimental verification and provided sufficient explanations for the effectiveness of the method, but appropriate adjustments need to be made in the order of chapters.

Cite this review as

Reviewer 2 ·

Basic reporting

This paper proposes a network named Local Spatial Projection Convolution, and provides a detailed description of the modules of local spatial projection ,anisotropic relative feature encoding and feature weight assignment.
Some suggestions for modification are as follows:

Experimental design

- It is necessary to analyze the data results of graphs and tables, which is relatively missing in this paper.Not only do graphic data displays but also data-based results are required
- Work on comparison experiments could be supplemented. Add comparative algorithms. Evaluate the effectiveness of the proposed methods when optimised individually and in combination.
- Some of the content in the method section should be moved to the evaluation section, and the experimental setup is missing from the paper.

Validity of the findings

None

Additional comments

- When introducing relevant work, choose those that are relevant to the methods used in your paper and detail the most relevant work, writing briefly about the others.
- The three distance functions in the feature weight assignment module lack a more detailed description, and Figure 4 lacks content keywords.

Cite this review as

---

## Round 0.2 · Minor Revisions

Dear authors,

The original Academic Editor is unavailable so I am taking over handling your submission.

Thank you for the revision. Currently, your article has a few remaining issues. We encourage you to address the concerns and criticisms of the reviewer and resubmit your article once you have updated it accordingly.

Best wishes,

Reviewer 1 ·

Basic reporting

The author has correctly responded to my concerns.

Experimental design

very well.

Validity of the findings

The author has correctly responded to my concerns.

Cite this review as

Reviewer 3 ·

Basic reporting

The paper presents a method for 3D point cloud deep learning that aims at exploiting local point features. The idea is to design a local spatial projection convolution operator, namely LSPConv, to process point cloud input. To address the anisotropy of point distribution, the authors proposed two modules namely Feature Weight Assignment and Anisotropic Relative Feature Encoding to improve the local feature learning by assigning weights to neighbor points and encoding points adaptively. The proposed method is tested with point cloud classification, part segmentation, and semantic segmentation task. The experiments show remarkable performance compared to previous methods.

The paper addresses a traditional but important topic in point cloud deep learning. However, this topic was new by 2017 when 3D deep learning was just emerged. As of 2023, there have been several methods on this topic, though.

The literature review covers most important papers in the area. Additionally, I also think the local feature learning for rotation invariant point clouds is also relevant to this work. It is considered a more challenging form of point cloud classification when the points are arbitrarily rotated in the 3D space.
[A] Zhang et al., Global Context Aware Convolutions for 3D Point Cloud Understanding, 3DV 2020.
[B] Zhang et al., RIConv++: Effective Rotation Invariant Convolutions for 3D Point Clouds Deep Learning, IJCV 2022.

Experimental design

There are a few technical points about which further explanation would be great.

1) The feature weight assignment module assigns weights to neighbor points based on three functions, but all of them appear to be fixed and not learnable. Have the authors considered using learnable parameters for the radial basis function?

2) In Equation 5, is there any chance that ReLU returns zero? It would be good to address edge cases to avoid division by zero in Equation 4.

3) It is unclear the ablation study is conducted on which task. Based on the metric it seems to be a part segmentation task. I wonder why this task was chosen for conducting the ablation study. Why do the authors not choose the classification task which is easier and faster to perform with more datasets available? Regarding the presentation of this ablation study, as the authors decide to present ablation studies before the main results, it would be great to explain about the task used for ablation study, its dataset, etc.

Validity of the findings

The experiments presented in the paper are somewhat satisfactory. There remain some issues that I have concerns about, as shown below.

1) The experiments show the effectiveness of the proposed modules with ablation studies demonstrating improvement for the part segmentation task. However, I found that the results do not fully report comparison with some recent methods, e.g., PointNeXt on the part segmentation and semantic segmentation task. I can see that the comparison of PointNeXt for classification task is reported, though.
[C] Qian et al., PointNeXt: Revisiting PointNet++ with Improved Training and Scaling Strategies, NeurIPS 2022.

2) The classification task also lacks an experiment with real-world data. Testing with ScanObjectNN would be great to measure the classification for such dataset.

Additional comments

The presentation of the paper is at a readable level for someone who is familiar with this topic. There remain some clarity issues that it would be great to fix. For example, I found that the intuition is not sufficiently explained especially in the Method section. Figure 1 and its caption focus on explaining the process by describing that module A is going to call module B, etc. Such a description is lengthy and will make readers difficult to track the intuition of the method.
This trend appears in other module descriptions as well, so it will be great to revise the writing for better clarity.

I also found some other minor issues with the presentation, which I listed below.

1) Repetitive sentences.

- PAConv (Xu et al., 2021) designed a Weight bank containing a set of weight matrices to map the features of the point cloud. This sentence appears in Line 135 and Line 142. In general, the description of PAConv in Section 2 should be significantly shortened.
- Again, the module names are repeatedly mentioned in the first paragraph of Method. Please consider rewriting this section.

2) Typo: Segmantation, Classifacation, isn’t adequately -> is not adequately

3) It will be good to simplify the math symbols in the equations. I found that some of them are unnecessarily complex, like the tilde and some accent symbols above the characters Equation 11, 12.

Cite this review as

---

## Round 0.3 · Minor Revisions

Dear authors,

The original reviewer was unavailable so I invited a new reviewer for final decision.

Thank you for the revision. Currently, your article has still a few remaining issues. We encourage you to address these minor concerns and criticisms and resubmit your article once you have updated it accordingly.

Best wishes,

Reviewer 2 ·

Basic reporting

1. There is no clear analysis of the motivation for using this model.
2. The introduction and review section lacks an analysis of the latest literature, such as 2023.

Experimental design

1. The experimental results lack a comparison of the latest methods.

Validity of the findings

1. It can be found that only 0.3% improvement is produced after using the method in the paper, which is difficult to say whether it is the advantage of the method or a random effect.

Additional comments

N/A

Cite this review as

---

## Round 0.4 · accepted · Accept

Dear authors,

Thank you for addressing all the reviewers' comments clearly. Your article is now acceptable for publication after the final revision.

Best wishes,